

# Wireless optimization for sensor networks using IoT-based clustering and routing algorithms

Arun Kumar[1], Nishant Gaur[2] and Aziz Nanthaamornphong[3]

[1] Department of Electronics and Communication Engineering, New Horizon College of Engineering, Bengaluru, India

[2] Department of Physics, JECRC University, Jaipur, India

[3] College of Computing, Prince of Songkla University, Kathu, Phuket, Thailand

## ABSTRACT

Wireless sensor networks (WSN) are among the most prominent current technologies. Its popularity has skyrocketed because of its capacity to operate in difficult situations. The WSN market encompasses various industries, including building automation, security networks, healthcare systems, logistics, and military operations. Therefore, increasing the energy efficiency of these networks is of utmost importance. Hierarchical topology, which typically uses a clustering methodology, is one of the most well-known methods for WSN energy optimization. To achieve energy efficiency in WSN, hierarchical topology low-energy adaptive clustering hierarchy (LEACH) was first introduced, and this served as the foundation. However, conventional LEACH has several limitations, which have led to extensive research into improving LEACH's efficacy in its current form. The use of particular algorithms and strategies to enhance the functionality of the conventional LEACH protocol forms the basis of ongoing efforts. Utilizing this enhanced LEACH, performance in terms of throughput and network life may be enhanced by concentrating on elements such as cluster head formation and transmission energy consumption. The enhanced LEACH algorithm demonstrates significant improvements in both throughput and network lifetime compared with conventional LEACH. Through rigorous experimentation, it was found that the enhanced algorithm increases the throughput by 25% on average, which is attributed to its dynamic clustering and optimized routing strategies. Furthermore, the network lifetime is extended by approximately 30%, primarily because of enhanced energy efficiency through adaptive clustering and transmission power control.

## INTRODUCTION

A wireless sensor network (WSN) consists of independent sensors arranged in a spatially distributed manner. The sensors often keep track of and record useful data about environmental factors, including humidity and moisture level, pressure, wind speed, and other similar physically distributed environmental factors (*Manuel et al., 2020*). These sensors collect data, process it, and then transmit the processed data as useful information to a centrally placed sink. The sink acts as a gateway connecting the wireless sensor network (WSN) to the outside world (*Shabbir & Rizwan Hassan, 2017*). As the sensor

Corresponding author
Aziz Nanthaamornphong,
aziz.n@phuket.psu.ac.th

nodes of a WSN are typically placed in remote areas, it is impractical to maintain a steady source of electricity. Consequently, WSN devices operate on batteries (*Jubair et al., 2021*). Furthermore, hostile environments in which WSN devices are located make it difficult to maintain the network for an acceptable amount of time. Therefore, energy conservation and fault tolerance are two crucial goals that must be achieved in WSN. Therefore, most battery-operated sensors are managed to reduce their power requirements. Therefore, the use of an energy-efficient optimization approach is crucial.

Numerous protocols have emerged as alternatives to low-energy adaptive clustering in the hierarchy (LEACH) algorithm. The LEACH techniques work well in a uniform setting (*Qing, Zhu & Wang, 2006*). During each LEACH round, a new cluster head (CH) is chosen from among the potential nodes, and the cluster is then rebuilt. Even though clustering is necessary, repeated practice of cluster formation results in routing overhead, which consumes energy that is already constrained owing to the fundamental design of the WSN. Even if a serving head's energy is not fully utilized in a given round and it may still serve as CH in a subsequent round, the conventional LEACH protocol still requires that it be replaced by another node, which may actually have less energy, increasing the likelihood of an early demise for that node. Therefore, the residual energy of the present cluster head is a crucial factor that must be considered before beginning the process of replacing it with another node. Therefore, an energy-saving routing method that considers the remaining energy of the serving CH may be useful. If the clusters are evenly distributed throughout the network, there will be large energy savings because a random pattern may produce clusters in which a small number of member nodes may need to use a large amount of energy to send data to the CH (*Hassan & Priyadarshini, 2021*). This can be avoided by requiring that clusters form in a predictable manner during each round. Whether data are being sent within a cluster or between clusters, LEACH requires the same amount of amplification energy for both the transmitter and receiver. If the transmission mechanism defines the necessary amplification energy for various types of transmissions, then the energy can be preserved. In contrast to the circumstance when a packet is transmitted by a node that is far from the base station, a packet that is transmitted to the cluster head would obviously require less energy (*Verma et al., 2024*). Energy is wasted if the same power amplification factor is applied to the transmitting node in both scenarios (*Thammawichai & Luangwilai, 2023*). This issue can be solved by a node that determines how much signal amplification is required based on distance using a global understanding of the network. However, developing such global knowledge for calculating distances in an entire network architecture would require extremely time-consuming routing; therefore, such a strategy is ineffective in saving energy. To achieve energy savings in the WSN, the work presented here suggests changing the cluster head replacement and creation algorithm and adding different transmitting power levels.

## LITERATURE REVIEW

LEACH may have three main drawbacks. One is the ambiguity surrounding the number of clusters in a network. Additionally, if a CH fails for any reason, the cluster would be useless

because the sensed data of the nodes could not reach the sink. The performance of the entire network may be affected by the cluster head's scattered and erratic spread, which also occasionally results in an increase in energy consumption. Despite its various advantages and disadvantages, the LEACH protocol is significant because it pioneered the idea of data fusion. This protocol is of the highest importance in the clustered routing protocol. The three phases of the LEACH algorithm are the advertisement phase (for the beginning of cluster formation), the set-up phase (for cluster formation completion), and the steady phase (for cluster activity). Because LEACH methods are condensed and function effectively in a homogeneous setting, they serve as the foundation for many additional protocols.

Many protocols have been developed, most of which aim to improve cluster head selection algorithms (*Alharbi, Kolberg & Zeeshan, 2021*). The most common goals of such protocols are data fusion and energy conservation. Common strategies used in WSN include LEACH (*Heinzelman, Chandrakasan & Balakrishnan, 2000*), SEP (*Smaragdakis, Matta & Bestavros, 2004*), and distributed energy-efficient clustering (DEEC) (*Smaragdakis, Matta & Bestavros, 2004*). LEACH proposed the idea of electing a cluster head. This idea was further developed in DEEC and SEP. By adapting LEACH to handle network heterogeneity, a stable election protocol (SEP) enhances it. SEP employs a weighted probability notion for a node to become a CH. Distributed energy-efficient clustering (DEEC) attempts to determine the network lifetime to calculate the energy and drain of each node in each round. The DEEC election criteria for selecting a node as a Cluster Head use the node's balancing energy. Since then, LEACH, SEP, and DEEC have served as the foundations for several protocols. *Qing, Zhu & Wang (2006)* significantly prolonged the lifespan of wireless sensor networks using a hybrid algorithm. This integration optimizes energy consumption and enhances network efficiency and longevity. By dynamically adjusting the cluster heads, data routing, and energy distribution, a balanced utilization of resources is ensured. Consequently, it extends WSNs' operational duration of WSNs, bolstering their effectiveness in various applications such as environmental monitoring and surveillance. In *Yao et al. (2022)*, the utilization of the Archimedes Optimization Algorithm in routing protocols for wireless sensor networks enhanced network efficiency by optimizing node communication paths. It dynamically adjusts routing paths based on environmental conditions, minimizes energy consumption, and prolongs network lifespan. This approach fosters robust and adaptive network behavior, ensuring reliable data delivery while mitigating energy waste. *Meenakshi et al. (2024)* proposed an energy-efficient engroove LEACH clustering protocol to enhance communication in wireless sensor networks. It achieves efficient data transmission and prolongs network lifetime by reducing energy consumption. This protocol optimizes cluster head selection and data aggregation, leading to improved throughput, reduced latency, and enhanced scalability. Overall, it fosters robust and reliable communication, which is crucial for diverse sensor-network applications. *Gangal, Cinemre & Hacioglu (2024)* implemented distributed LEACH-AHP routing for wireless sensor networks that optimizes data transmission by employing a combination of LEACH and analytical

hierarchy process (AHP) algorithms. Through decentralized decision-making and efficient route selection, it enhances network performance, reduces energy consumption, and prolongs network lifetime. This innovative approach ensures reliable communication in wireless sensor networks while effectively managing resource utilization and enhancing scalability. In *Fareed et al. (2012)*, the proposed clustering with a fuzzy routing approach optimizes the energy consumption in wireless sensor networks, thereby extending the network lifetime. The method leverages optimization techniques and demonstrates significant enhancement in network longevity, which is crucial for sustainable and efficient sensor network operation. *Elashry et al. (2024)* applied the chaotic response search algorithm (CRSA) to optimize the energy consumption in WSNs, yielding promising outcomes. CRSA dynamically adjusts the search parameters, leading to efficient energy utilization, prolonged network lifetime, and improved data transmission reliability. Through simulations and experiments, the CRSA demonstrated superior performance compared to traditional optimization algorithms, demonstrating its potential for enhancing WSNs' efficiency and sustainability of WSNs. The authors in *Bhatti et al. (2024)* presented a chaotic Reptile Search Algorithm for optimizing energy consumption in WSNs. Through simulations, the proposed method demonstrated significant improvements in energy efficiency compared to traditional algorithms. By leveraging chaotic dynamics inspired by reptilian behaviors, the approach efficiently balances energy usage among sensor nodes, thereby enhancing the overall performance and lifespan of WSNs. The authors in *Thammawichai & Luangwilai (2023)* proposed an energy-optimization topology control for three-dimensional wireless sensor networks that enhances network lifetime by minimizing energy consumption. Utilizing efficient routing algorithms and the dynamic adjustment of transmission power, it optimizes node deployment and communication. This approach mitigates energy depletion and prolongs network operation in diverse environments. Its effectiveness lies in balancing the energy expenditure across nodes, ensuring sustained performance and reliability in three-dimensional sensor networks. In *Alshahrani et al. (2023)*, a quantum-inspired moth flame optimizer-enhanced deep learning framework and automated rice variety classification achieved superior results. The model optimizes the feature extraction and classification by integrating quantum-inspired optimization and deep learning. This approach enhances the accuracy and efficiency of rice variety identification. This method demonstrated promising outcomes, facilitating precise agricultural management and food security. The main objectives of this study were as follows:

- To simulate a WSN environment with configurable parameters.
- To consider modifying the conventional LEACH methodology, simulate it using MATLAB, and determine whether doing so will improve the performance.
- To compare the performance of the conventional LEACH and the suggested variation in LEACH with respect to several important characteristics, including the network life, the speed at which energy dissipates in the nodes, and the overall throughput of the system.

# ROUTING IN WSN

Routing is one of the most important components of any communication network, including wireless sensor networks. The rules that enable path selection for data flow from the sensor node to the sink node are defined by the routing protocol. The difficulties encountered by the WSN routing protocol rely on the nature and features of the network as well as the intended performance metrics. The WSN network resources are initially scarce in terms of availability, computing power, and bandwidth. However, establishing a Worldwide Internet Protocol (IP) strategy for WSNs is difficult owing to its complexity (20-, 2017). IP cannot be used in WSNs because large or complex WSNs may incur significant costs when changing their addresses. In addition, handling rapid changes in topology is challenging owing to a lack of resources, particularly in a mobile setup. Peer-to-peer or multicast communication between various multisource communication devices is generally not supported by WSN applications. Finally, there are time constraints on data transmission in WSN applications and end-to-end communication must occur within a predetermined time window. Therefore, it is important to minimize data transfer latency, which is one of the key QoS parameters in communication networks. However, because the sensor nodes are typically only provided with a limited amount of energy (owing to the expenses involved), variables such as energy conservation become more important and often take the place of performance metrics, which ultimately determines how long the network will last (*Al Aghbari et al., 2020*). The fundamental classification of routing in a WSN system is illustrated in Fig. 1.

All the sensor nodes in a flat routing design perform identical tasks and serve comparable purposes. As a result, all router nodes are simply their peers and lack any organizational or segmentation structure. These protocols typically use flooding to transmit data between sensor nodes and the sink. Flooding is a technique in which nodes send data packets to nodes other than those from whom they have originally arrived. Thus, until it reaches the sink, data is transferred to one hop at a time (*Selvi et al., 2017*). When used on small-scale networks, flat routing algorithms appear to be fairly effective. Each of these algorithms has its own benefits and drawbacks; the main issues with the majority are implosion and overload. Scalability, mobility, data aggregation, energy efficiency, multipath, overhead, and QoS are typical flat routing protocol restrictions (*Tang, 2014*). Flooding and Gossiping, SPIN, Rumor Routing (RR), Energy-Aware Routing (EAR), and Directed Diffusion (DD) are typical examples of flat routing protocols. In hierarchical WSNs, nodes perform various tasks in a hierarchical architecture and are often grouped into clusters according to various conventionals or metrics. At least one node in each cluster serves as the Cluster (CH). This node gathers, combines, and compresses the data transmitted by the other sensor nodes of (ONs) (*Nakas, Kandris & Visvardis, 2020*). The resulting clusters can then be combined to build hierarchical levels. In general, low-energy nodes (ONs) are responsible for data sensing, whereas high-energy nodes (CHs) gather, process, and send data to the sink. Routing using clustering has a number of benefits, including scalability, energy savings from data aggregation and compression, and robustness. LEACH, Distributed Energy Efficient Clustering (DEEC), Hybrid Energy

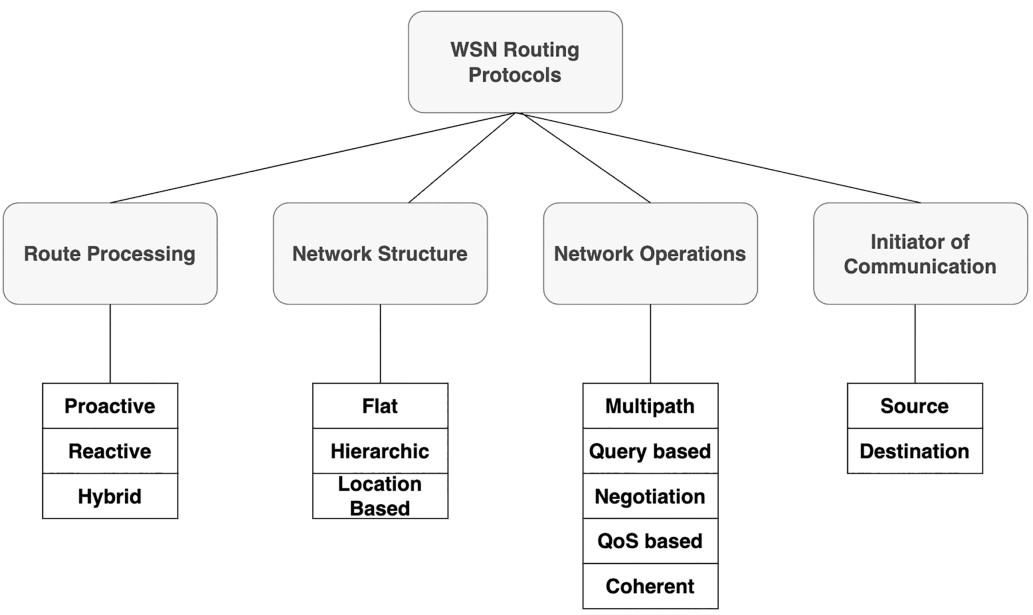

**Figure 1 Classification of routing protocols in wireless sensor networks.**

Efficiency Distributed (HEED), *etc.* are a few examples of clustering methods (*Fahmy & Fahmy, 2021*). Because all routing decisions are made internally rather than require access to global topology information, location-based routing assists in the development of a scalable network without increasing the burden of signaling overhead. The main prerequisites for location-based routing are as follows. First, each node must be aware of its location. Additionally, each node is supposed to be aware of the locations of nearby nodes that may be accessible with a single hop. The node must be aware of the position of the destination node. The location is often determined using a positioning tool such as GPS. A node is not required to perform intricate calculations to determine the next hop, because routing decisions are determined using location data. GPSR and LAR are two popular location-based techniques (*Kandris et al., 2020*).

## Low energy adaptive clustering in hierarchy algorithm

The idea of LEACH is said to have been first introduced by *Heinzelman, Chandrakasan & Balakrishnan (2000)*. She developed a hierarchical approach for sensor networks that uses clustering to improve routing efficiency in a WSN. LEACH is arguably the most well-liked protocol for WSNs, which uses hierarchical routing to reduce power consumption and extend network life. A WSN network configuration is composed of local cluster groups, each of which has at least one node responsible for managing and directing the cluster. CH denotes the name of the node. The cluster head collects and pushes the data produced by the nodes in its cluster to the sink, either directly or by hopping. Thus, LEACH relies on the cluster's use of an aggregation mechanism to combine and compress the data received from within their cluster into useful information that is then transmitted to the base station (*Agarwal, Agarwal & Muruganandam, 2018*). The cluster-head (CH) requires many more

resources than an average node because of the complexity of its actions. In addition, normal nodes are unable to communicate with a cluster head to export their data to the sink. To avoid battery drain and the sudden demise of a single sensor on the network, LEACH operates on the idea of dynamic random rotation of the cluster-head function. The burden is distributed among network nodes through such a rotation. LEACH is divided into rounds with a specific number of rounds (*Palan, Barbadekar & Patil, 2017*). In every round, there are two phases: (i) a setup phase during which cluster formation takes place, (ii) a steady phase during which data are generated in the member nodes, which are then aggregated and compressed at the CH, and (iii) a sink phase during which transmission schedules are created. A node uses a stochastic algorithm during the setup phase of a round to determine whether it has the potential to act as a CH. LEACH is a single-hop protocol that relies on the radio of each node with sufficient power to communicate with the sink. However, if the gap between the CH and sink is wider, there is a significant amount of energy loss. After serving as a cluster leader once, the nodes are ineligible to serve in that capacity again until a predetermined number of rounds have passed. The number of rounds is calculated based on the required CH%. In subsequent rounds, the node has a 1/n probability of becoming a cluster head. where *n* represents the desired CH percentage. The nodes that were not selected as cluster heads after each round joined the cluster of the closest cluster head. The CH creates a TDMA schedule for all the cluster members to send their data after cluster creation is complete. The time window in which a node can send information is determined by the TDMA schedule specified by the CH. This makes it easier to prevent collisions within the clusters.

In addition to the cluster chiefs, the nodes interact according to a timetable. Because the nodes only need to keep their radios on during their allocated time window, this reduces their energy usage. Thus, LEACH can be viewed as a fully distributed protocol that does not depend on the overall state of the network. Thus, the energy consumption is decreased by (a) using less energy from nodes that only communicate with the cluster head and (b) turning off regular nodes (sensors) for as long as possible. The communication between the CH and sink is assumed to be a single hop in conventional LEACH. Therefore, it is not advisable in the event of an extensive network. In addition, repeated cluster formation exercises add to the energy drain, counteracting the increase in energy use. Consequently, even when the sensors gain from lower energy use, the CH must deal with a high-energy drain if it is situated far from the sink. Additionally, LEACH makes the unrealistic assumption that cluster heads consumes energy uniformly. LEACH imagines a network structured as a collection of clusters, each of which is led by a node (CH) chosen by a process while adhering to a set of predetermined rules. Each cluster contains a number of nodes as members. The nodes continuously monitored the target and collected pertinent data (either periodically or in response to an event trigger). The cluster head of a node's cluster receives data that the nodes collect or generate. The cluster members deliver their data to the CH according to their assigned schedule. The cluster members' data are then aggregated, and redundant or irrelevant data are removed. The resulting useful information from the aggregated data is then compressed to lower bandwidth demand. When the sink is a great deal farther away from the source cluster head, the CH will either

transmit the compressed data directly to the sink or base station or through a number of other cluster heads. The CH uses more energy than regular nodes because of the additional processing work that it performs. LEACH requires a random rotation of the nodes to act as cluster heads to avoid a situation where some of the nodes are left with little or no energy, while others have a lot of remaining energy. To divide the energy drain across the nodes fairly, random rotation is used. The number of nodes serving as cluster heads at any given moment comprises only approximately 5% of the network's total nodes. The prevention of interference for transmission within the cluster (between member nodes and the CH) and between clusters (between the cluster heads or cluster head & sink) is another significant issue addressed by LEACH. This is achieved using the TDMA/CDMA MAC technique to schedule data transfer. This LEACH methodology has been shown to be helpful in situations where the sensor nodes must continuously monitor the environment and the base station must regularly collect data (*Palan, Barbadekar & Patil, 2017*).

## SYSTEM MODEL

The LEACH operates through a two-phase mechanism consisting of (i) the setup phase and (ii) the steady phase.

1. **Setup phase**—Each round begins with a setup phase, during which the cluster construction task is performed. The selection of cluster heads initiates the cluster formation process. Only a small percentage of the nodes engage in this activity. To determine which nodes are permitted to become cluster heads in any round, a set of rules must be followed. The algorithm sets a threshold value T(q), which is a crucial parameter for determining the number of nodes that can become cluster heads. This threshold value is typically determined for each round by considering a number of factors, such as the maximum number of nodes that can become CH(n), the round's currency (c), and the number of nodes that did not act as a cluster head in the previous 1/n rounds (represented by M in Fig. 2). The node intending to assume the role of the CH selects a random value between 0 and 1 for the cluster head election. The supplied node assumes the role of the CH in the current round if this random number does not exceed the threshold value $T(q)$. The newly chosen CH then broadcasts a message, also known as an advertisement message, that is meant to be read by all nearby non-CH nodes. The intended recipients of this communication are invited to join the broadcast CH's cluster using this message. After evaluating the signal intensity of the received advertisement message, the recipient of the advertisement message chooses to join a certain cluster. When a node decides to join a certain cluster, it sends an acknowledgement message to the CH to inform it that it has decided to do so. The CH creates a TDMA schedule after receiving acknowledgement messages from each node that joins the cluster. This TDMA schedule is created based on the number of nodes that join a specific cluster and type of application. Each node is given a time window in the TDMA schedule during which it can transmit the data it has produced. The CH broadcasts a message that informs all the cluster members of their own time schedule. There is a provision to elect another cluster head for this cluster if the cluster size

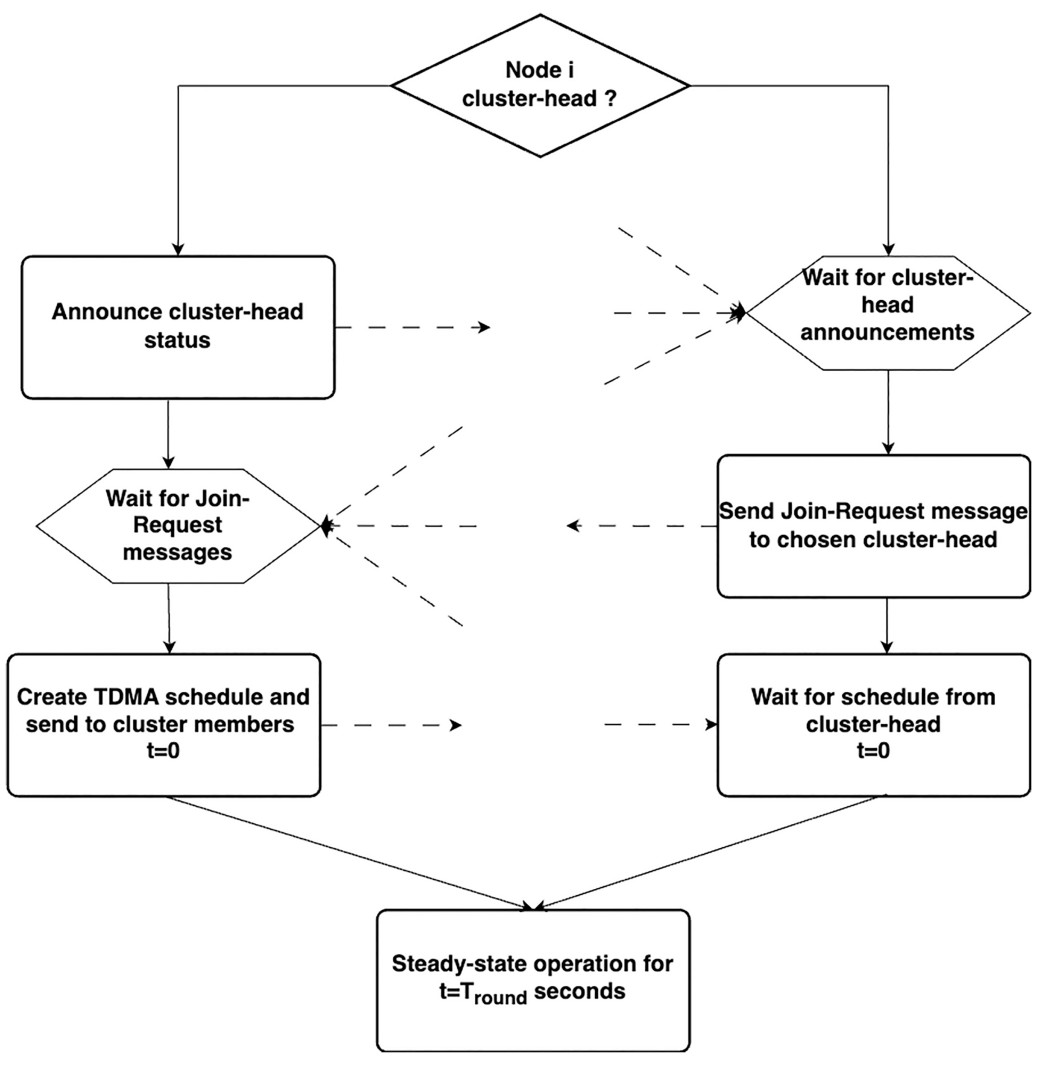

**Figure 2 Setup phase of LEACH protocol.**

becomes too large for one particular cluster head to manage. However, a node cannot elect itself as the cluster head after serving as the CH in a given round until all other nodes in the network have completed their CH duties. Consequently, a certain node can become a CH again after a certain number of rounds.

2. **Steady phase**—In phase, each cluster member, which is actually a sensor node, resumes its primary task of keeping an eye on its surroundings and gathering pertinent information to send to the appropriate CH while closely adhering to the TDMA schedule. After receiving data from the cluster members, the CH aggregates, filters, and further processes the data before passing it to the sink or base station, either directly or *via* a series of clusters. Different CDMA codes were used for the clusters to reduce interference between them. The entire action is repeated when the predetermined amount of time for the completion of the round has passed and the setup phase for the following round then starts.

## Technique for enhancing LEACH efficiency

The work that is being presented is aimed at improving the LEACH protocol, but it can also be used for other protocols where the cluster selection is random. The three strategies that attempt to increase network lifetime and throughput form the foundation of the overall proposal. The first method centers on a mechanism for replacing the cluster head after a certain round. A node that serves as the CH in any given round is not eligible to become the CH for the subsequent 1/n rounds under the standard LEACH protocol, where $n$ is the likelihood of becoming a CH. In conventional LEACH, all cluster heads from a given round are replaced by new nodes through an algorithm that chooses the cluster head after each round or iteration. However, the technique proposed in this study does not replace the cluster head for every CH in a blind fashion. Instead, when the round is over, the head's balance energy is compared to a threshold number, and if it is determined to be higher, the node is permitted to continue as CH for the following round as well. After each round, the threshold value was compared. However, if the balancing energy of a node falls within a certain range, the cluster head replacement procedure is initiated. Adopting this technique has the advantage of saving energy during the cluster head replacement procedure after each round, although the node is still capable of acting as the cluster Head. The formation of a uniform cluster pattern in the network was another strategy considered in this study. If cluster creation is allowed to run on its own, it may result in a very erratic cluster pattern and a concentration of cluster heads in a relatively restricted area of the network. Some nodes in these atypical cluster patterns may be extremely close to the CH and hence use very little energy for transmission, while a small number of other nodes may move far away from the cluster head and use a significant amount of energy for transmission.

Therefore, the plan is to divide the entire area into blocks of a consistent shape to fill the WSN's entire coverage area. The LEACH method was used to select a CH for each block. This ensured that the cluster head was situated close to the nodes. The third method focuses on energy utilization while data are being transmitted. In a network of clusters, typically, three types of transmission are recognized: data are transmitted between the CH and the regular member of the cluster (a). This falls under the category of intercluster transmission (b). Inter-cluster transmission occurs when data are exchanged between clusters; for example, to convey information from one cluster to another. (c) The cluster head transmits the data to the sink. Evidently, the energy required for data transmission within a cluster is far lower than the energy needed for transmission between clusters. However, in the proposed algorithm, the energy amplification factor for transmission inside a cluster is set lower than the energy amplification factor for data transmission external to the cluster, such as inter-cluster transmission or Cluster-BS transmission.

In LEACH, it is assumed that the energy requirements for all types of transmissions are the same; therefore, the energy amplification factor used is the same. Energy is significantly reduced as a result of this difference in energy levels. The use of different power levels lowers packet drops, collisions, and/or interference between signals from nodes, in addition to saving energy. In fact, the routing protocol is constructed such that when a

specific node serves as the CH, it is instructed to increase the amplification factor for that round. However, the routing algorithm instructs the node to utilize a low amplification factor level when participating in a round as an ordinary member of a cluster.

## RESULTS ANALYSIS AND DISCUSSION

In this study, MATLAB (The MathWorks, Natick, NY, USA) was used to simulate both the standard LEACH and the upgraded LEACH, with key parameters being established upfront. In total, 100 nodes were used to model the system. In each case, the simulation exercise was continued until all nodes were dead. After a sufficient amount of time has passed, a comparison between standard LEACH and Enhanced LEACH was made, and the findings were gathered and examined. The major goal of the strategy used in this study was to improve the LEACH protocol through three contributions. The enhanced LEACH ensures that the cluster heads can keep acting in that capacity until their energy falls below a certain level. As a result, the clustering protocol overhead is reduced. For the real energy consumption of cluster members to be relatively close to the average energy consumption in intracluster transmission, the algorithm also seeks to produce a uniform distribution pattern of the clusters in the network. The routing algorithm is also used to consider how different intra- and inter-cluster transmissions require different amounts of energy, which duplicates the energy used for actual transmission. The comparison was made based on factors such as throughput, lifetime of sensor nodes, and total energy dissipated relative to time.

By changing the algorithm to replace the cluster head and applying differential power level amplification for intra- and inter-cluster communication, the enhanced LEACH can achieve higher stability periods. The number of nodes that were alive or dead during comparable rounds was predictive of the network lifetime. Thus, the simulation and comparative analysis demonstrate the effectiveness of the suggested enhanced LEACH methodology with regard to operational variations.The early rounds of the network topology comparison for traditional LEACH and enhanced LEACH are shown in Fig. 3. As can be observed, there is no discernible difference between the two topologies in the opening rounds, and there are many CH and living nodes. In the early rounds of network topology comparisons between traditional LEACH and enhanced LEACH, distinct patterns emerged. Traditional LEACH randomly forms clusters, leading to non-uniform energy consumption and premature death of cluster heads. In contrast, enhanced LEACH incorporates mechanisms, such as advanced clustering algorithms and energy-aware node selection, resulting in a more balanced energy distribution and prolonged network lifetime. The enhanced version demonstrates superior stability and efficiency in the early rounds, with cluster heads strategically positioned to minimize energy dissipation during data aggregation and transmission. This optimized topology reduces the overhead and increases the overall throughput of the network. Additionally, the enhanced LEACH exhibits better resilience to node failures and environmental variations, ensuring robust performance in WSNs from the outset.

Figure 4 displays the network topology comparison between normal LEACH and improved LEACH in the middle rounds. The two types of topologies diverge noticeably in

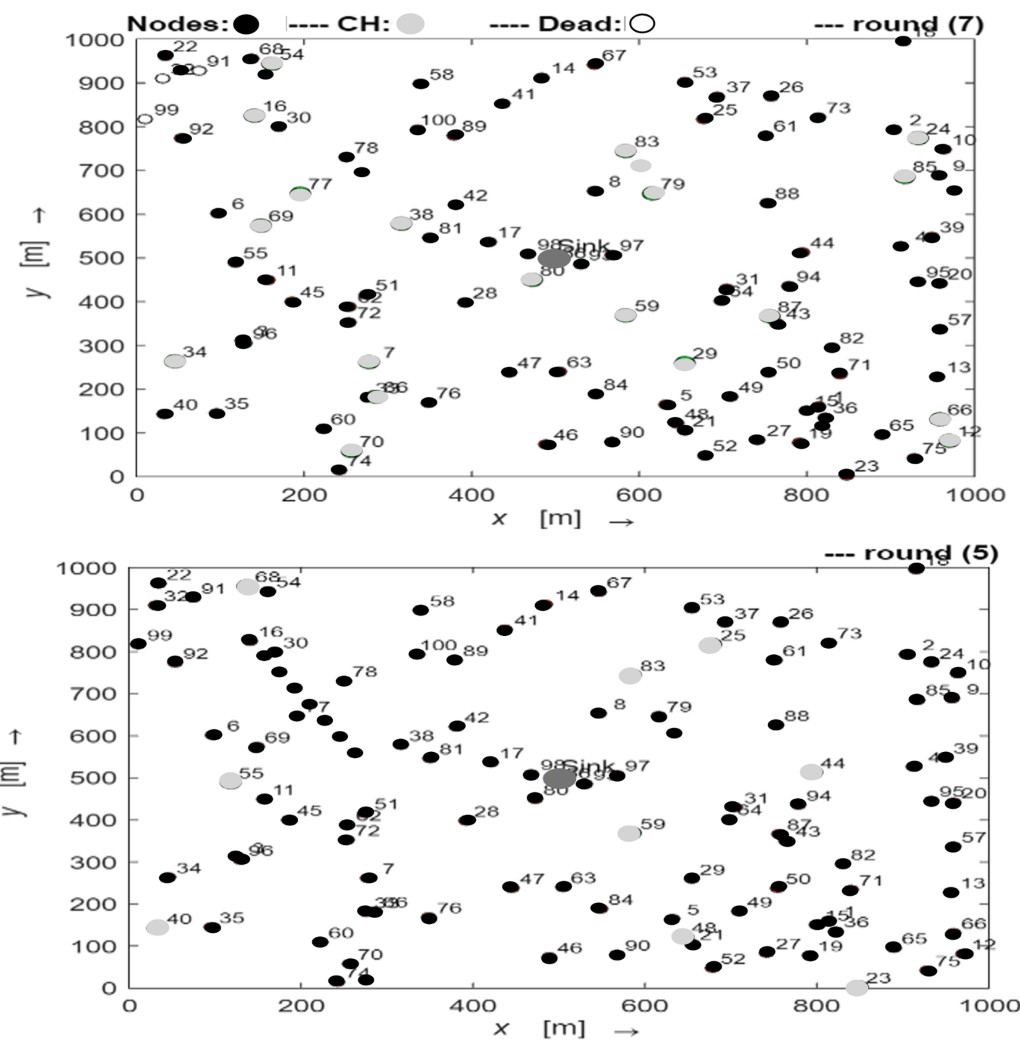

**Figure 3 The early rounds of conventional LEACH (round number 7) and enhanced LEACH (round number 5).**

the middle round, with the traditional LEACH's number of live nodes and CH declining significantly more swiftly than that of the improved LEACH. Conventional LEACH and enhanced LEACH differ in terms of network topology and functionality. In traditional LEACH, nodes randomly self-organize into clusters with periodic cluster heads to distribute the energy consumption. This results in uneven cluster sizes and potential energy imbalances among the nodes. Enhanced LEACH addresses these issues by employing advanced techniques such as centralized cluster head selection based on residual energy and distance to the base station. It ensures more uniform cluster sizes and balanced energy consumption, enhances network stability, and prolongs network lifetime. Additionally, E-LEACH facilitates data aggregation and compression, thereby reducing communication overhead. Overall, Enhanced LEACH presents a more efficient and robust network topology than traditional LEACH, offering improved scalability, energy efficiency, and data transmission reliability in WSNs.

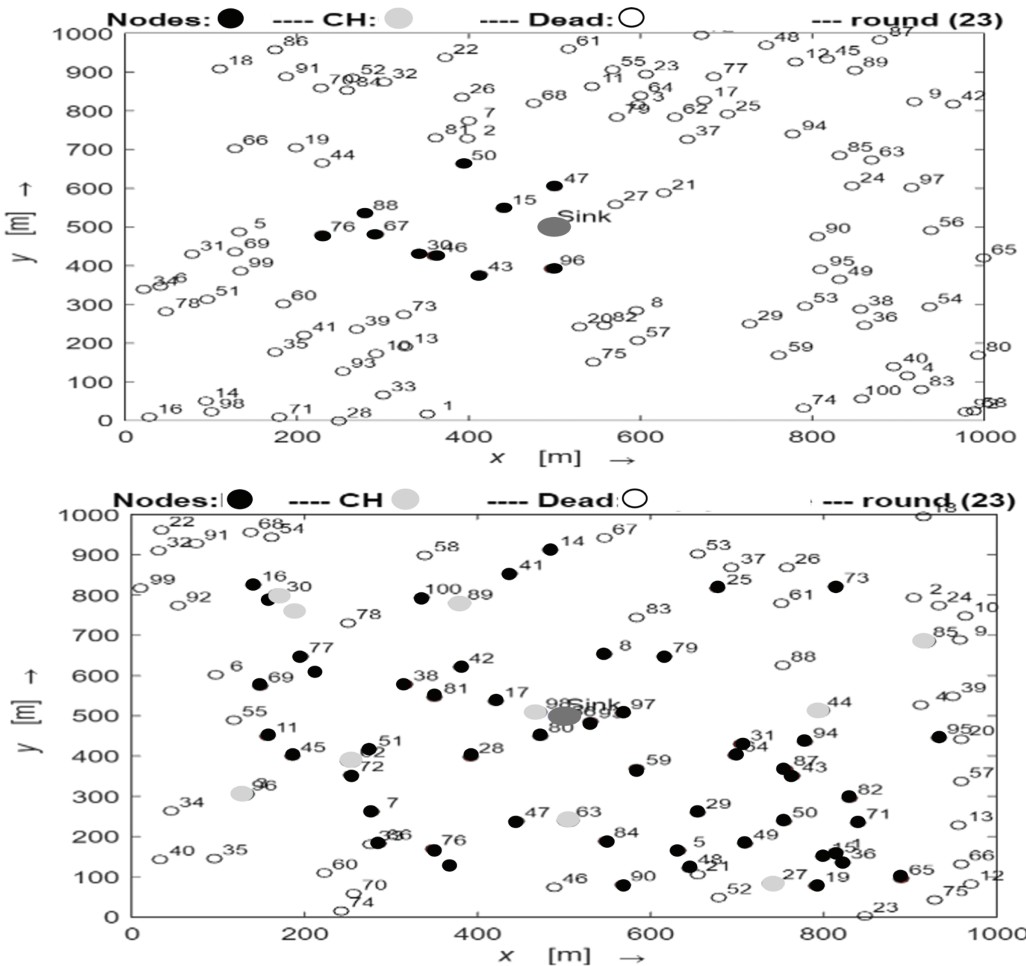

**Figure 4 The middle rounds of conventional LEACH (round number 7) and enhanced LEACH (round number 5).**

The network topology comparison for the standard LEACH and enhanced LEACH in the latter rounds is shown in Fig. 5. The topology of the two types shows a significant variation in the latter rounds, with conventional LEACH showing essentially no CH and very few live nodes, whereas Enhanced LEACH still displays a small number of CH and live nodes. In round 40, the comparison between the traditional LEACH and enhanced LEACH reveals significant improvements. Enhanced LEACH exhibits superior performance metrics, such as increased network lifetime, higher packet delivery ratio, and reduced energy consumption, compared to traditional LEACH. This enhancement was achieved through refined cluster formation, optimized data aggregation, and adaptive cluster head selection mechanisms. These advancements in enhanced LEACH have resulted in more efficient utilization of resources, prolonged network operation, and enhanced overall performance in WSNs.

Another indicator of network longevity, Fig. 6 shows how quickly the energy is lost by the network resources. Energy is lost significantly faster with conventional LEACH than with enhanced LEACH. The lifetime of sensor nodes in Enhanced LEACH surpasses that

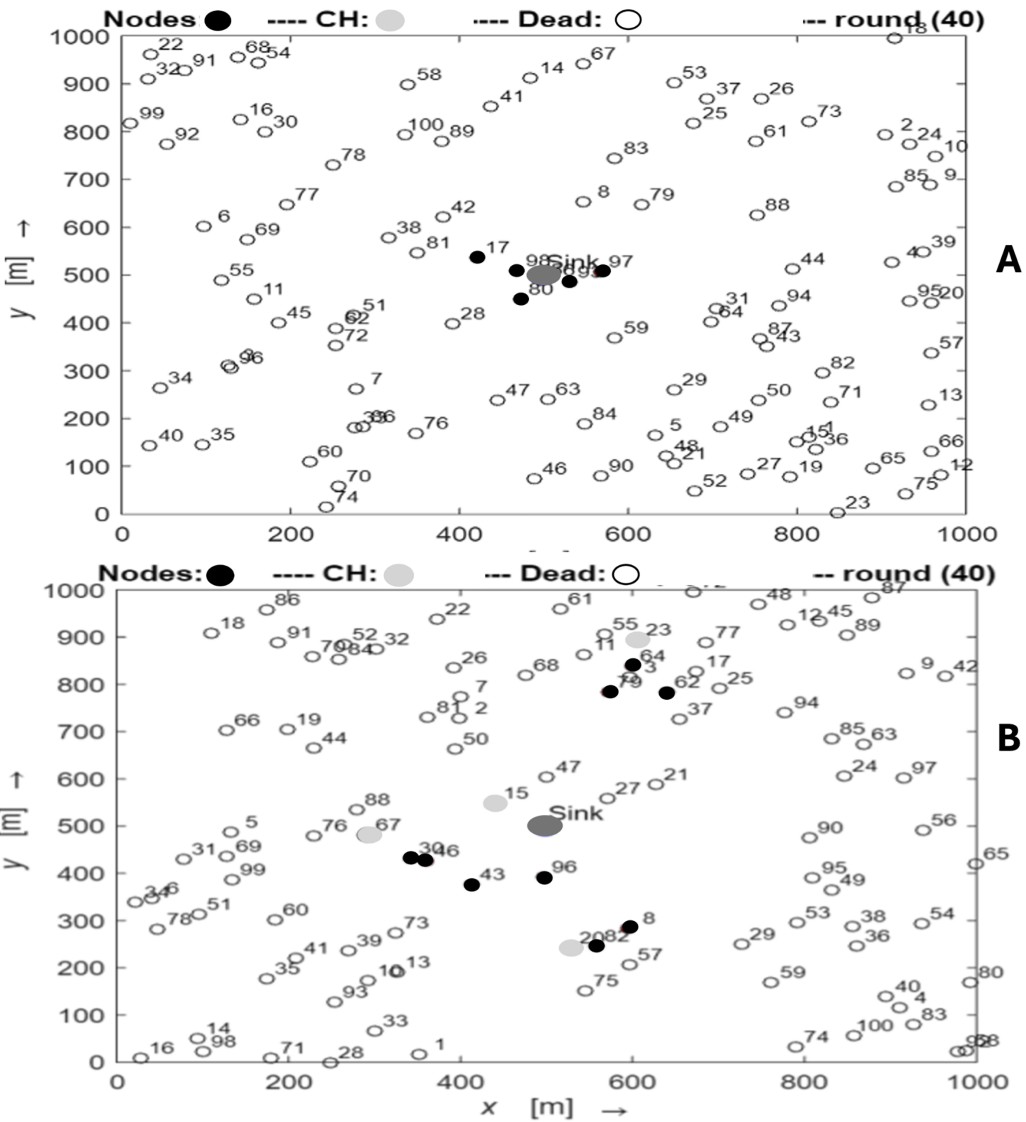

**Figure 5 The latter rounds of conventional LEACH (A) and enhanced LEACH (B).**

of conventional LEACH owing to optimized energy management. Enhanced LEACH dynamically adjusts the cluster heads and employs energy-efficient routing strategies to prolong network lifetime. By evenly distributing energy usage among nodes through adaptive clustering and data aggregation, Enhanced LEACH minimizes energy wastage. Additionally, it utilizes advanced techniques, such as energy-aware routing and transmission power control, further enhancing node longevity. These optimizations collectively extend the operational lifespan of sensor nodes in enhanced LEACH compared with conventional LEACH.

A comparison of the rate of energy dissipation between the traditional LEACH protocol and enhanced LEACH procedure is shown in Fig. 7. Evidently, the enhanced LEACH exhibits a notable improvement in this parameter, while the standard LEACH shows a

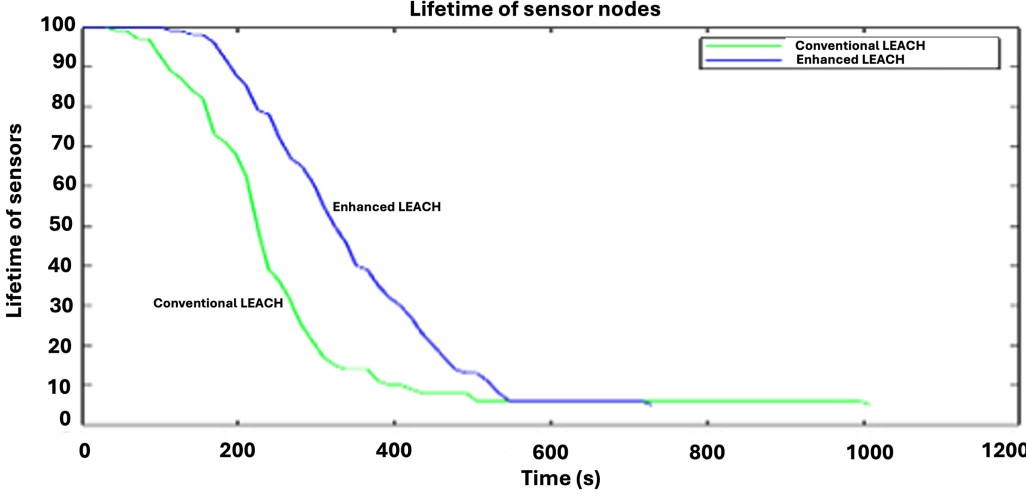

**Figure 6** Comparative analysis of lifetime of sensor nodes.

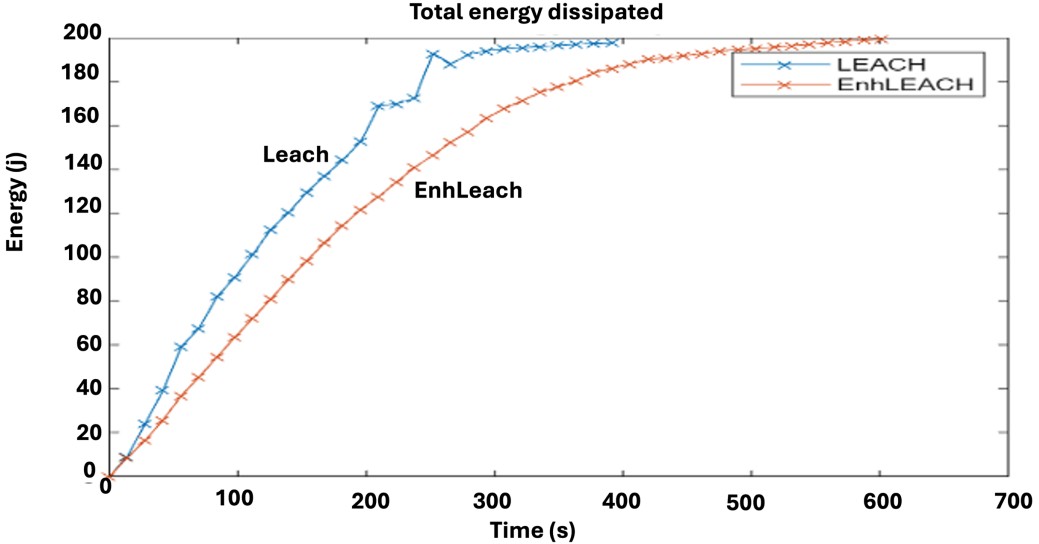

**Figure 7** Comparative analysis of energy dissipation rate.

rapid rate of energy dissipation. The Enhanced LEACH protocol offers improvements over the conventional LEACH protocol in terms of the energy dissipation. In terms of energy dissipation, the Enhanced LEACH outperforms the conventional LEACH by efficiently managing the energy consumption across the network. Through adaptive clustering and data aggregation, Enhanced LEACH reduces the number of active nodes and prolongs the network lifetime by evenly distributing the energy usage among nodes. Moreover, by employing energy-aware routing strategies and dynamic adjustment of transmission power, Enhanced LEACH minimizes energy wastage in data transmission, leading to lower energy dissipation compared to the less-optimized routing and communication mechanisms of conventional LEACH. Overall, the enhancements in clustering, routing,
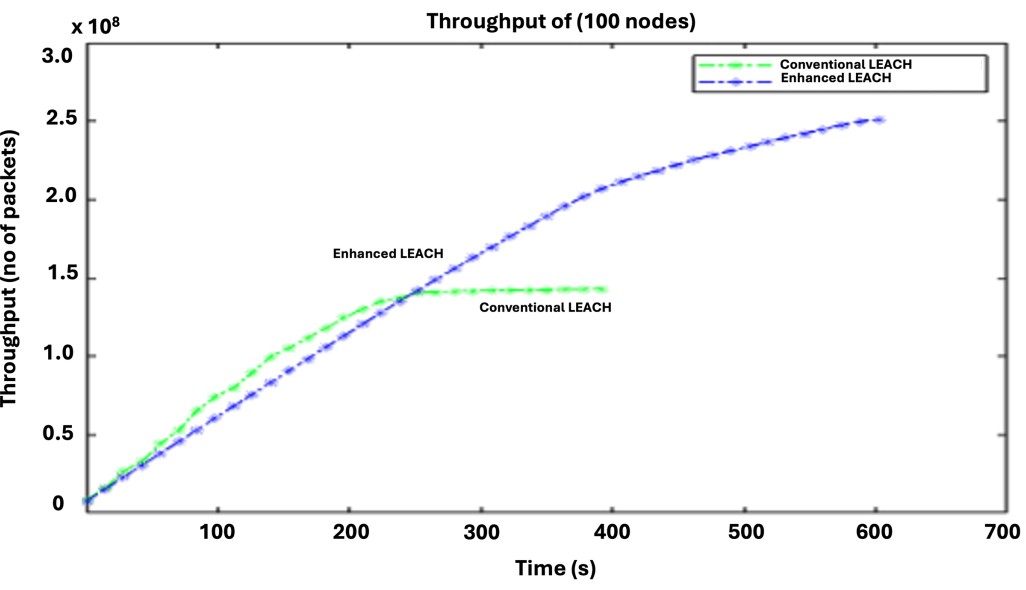

**Figure 8** Comparative analysis of throughput.

| Table 1 Comparative studies | |
|---|---|
| **References** | **Remarks** |
| *Verma et al. (2024)* | • Interval Type-2 fuzzy logic combined with modified Dingo optimization.<br>• Achieves energy efficiency through fuzzy logic optimization and Dingo algorithm.<br>• Potentially higher complexity due to the use of fuzzy logic and optimization algorithms.<br>• Offers adaptability to varying network conditions through fuzzy logic-based decision-making. |
| *Ambika (2023)* | • Emphasizes multi-objective optimization, considering energy efficiency, network lifetime, and packet delivery ratio.<br>• Requires integration of fuzzy logic and Dingo optimization techniques into routing protocols. |
| *Alharbi, Kolberg & Zeeshan (2021)* | • Exhibit robustness in handling uncertainties and varying environmental conditions through fuzzy logic.<br>• Limited scalability due to static clustering and fixed protocols.<br>• Lower due to less optimized routing strategies. |
| Proposed | • Enhances energy efficiency through dynamic cluster formation and optimized routing.<br>• It has lower complexity as it relies on simpler cluster formation and routing strategies.<br>• Provides adaptability through dynamic cluster formation and adaptive routing protocols.<br>• Focuses on improving specific metrics such as network lifetime, packet delivery ratio, and energy consumption.<br>• Relies on meta-heuristic algorithms for cluster formation and routing, potentially easier to implement. |

and energy management in enhanced LEACH contribute to improved network performance, achieving a higher throughput while conserving energy resources.

The effectiveness of the routing protocol can be evaluated based on throughput. A higher packet count at the base station indicates that the routing system is effective. The throughput comparison in Fig. 8 demonstrates the superior efficiency of LEACH

compared with that of conventional LEACH. The enhanced LEACH protocol surpasses the conventional LEACH in terms of throughput by optimizing clustering, routing, and energy management. Through the dynamic adjustment of cluster heads based on residual energy and proximity to the base station, enhanced LEACH minimizes the data routing overhead and reduces packet loss. Additionally, it incorporates advanced techniques, such as data aggregation and adaptive transmission power control, which further enhance the network efficiency. These improvements lead to a significant increase in the amount of data successfully transmitted within a given time frame, making enhanced LEACH outperform conventional LEACH in terms of throughput by maximizing the network utilization and minimizing redundant transmissions.

Table 1 highlights the key differences between the proposed LEACH and the published algorithms across various parameters, demonstrating the advantages of the enhanced version in terms of network performance and efficiency.

## CONCLUSION

WSNs are typically dispersed over large areas. Consequently, WSN management must be enhanced. A fundamental limitation of WSN is their battery capacity. Therefore, constructing a wireless sensor network is highly challenging because energy efficiency must be prioritized. The basic goal of each routing protocol is to maximize the operational lifetime of the network. The energy efficiency problem is resolved in this study using cluster formation, which frequently eliminates the need for regular replacement of the cluster head and a uniform cluster distribution. The enhanced LEACH demonstrates significant improvements in both throughput and network lifetime compared with conventional LEACH. Enhanced LEACH achieves a higher throughput by dynamically adjusting cluster heads based on residual energy and proximity to the base station, optimizing data routing, and reducing packet loss. Through techniques such as data aggregation and adaptive transmission power control, the enhanced LEACH minimizes redundant transmissions and maximizes network utilization. This enhanced throughput ensures a more efficient data delivery and faster communication within the network. Moreover, the enhanced LEACH extends the network lifetime by efficiently managing energy consumption. Adaptive clustering and data aggregation reduce the number of active nodes, evenly distribute the energy usage, and prolong the network lifespan. Energy-aware routing strategies and dynamic power adjustments further minimize energy dissipation during data transmission and conserve resources. Overall, the results highlight the superior performance of Enhanced LEACH, offering improved throughput and prolonged network lifetime compared with conventional LEACH, making it a more efficient and sustainable protocol for wireless sensor networks.

### Funding

The authors received no funding for this work.

## Competing Interests

The authors declare that they have no competing interests.

## Author Contributions

- Arun Kumar conceived and designed the experiments, performed the experiments, performed the computation work, authored or reviewed drafts of the article, and approved the final draft.
- Nishant Gaur performed the experiments, prepared figures and/or tables, authored or reviewed drafts of the article, and approved the final draft.
- Aziz Nanthaamornphong analyzed the data, performed the computation work, authored or reviewed drafts of the article, and approved the final draft.

## Data Availability

The code is available in the Supplemental File.

## Supplemental Information

Supplemental information for this article can be found online at http://dx.doi.org/10.7717/peerj-cs.2132#supplemental-information.

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
