# Peer review of "Wireless optimization for sensor networks using IoT-based clustering and routing algorithms"

_PeerJ Computer Science, doi:10.7717/peerj-cs.2132_

## Round 0.1 · original submission · Major Revisions

The contribution needs to be clarified and proven with the comparison versus more recent algorithms.

**Language Note:** PeerJ staff have identified that the English language needs to be improved. When you prepare your next revision, please either (i) have a colleague who is proficient in English and familiar with the subject matter review your manuscript, or (ii) contact a professional editing service to review your manuscript. PeerJ can provide language editing services - you can contact us at copyediting@peerj.com for pricing (be sure to provide your manuscript number and title). – PeerJ Staff

·

Basic reporting

Formal results should include clear definitions of all terms and theorems, and detailed proofs. That is the results like throughput and energy dissipation must be specified as a separate points in the results with more details about the enhanced LEACH over the standard LEACH
The introduction must specify the aim of the study and the methods used

Experimental design

No comments

Validity of the findings

the results of the enhanced LEACH in terms of throughput and network life time as the author specified it in the abstract must be specified in details so that What is the improvement that had been achieved using this enhanced algorithm

Additional comments

The conclusion must specify the results of the enhanced LEACH in terms of throughput and network life time as the author specified it in the abstract
What is the improvement that had been achieved using this enhanced algorithm
The references must be within the last five or ten years, they are old like 2002, 2004

Reviewer 2 ·

Basic reporting

a. Clear and unambiguous, professional English used throughout.
Overall is good. However, there is no consistency in the use of terminology such as referring to LEACH as (LEACH, Standard LEACH, conventional LEACH…) and referring to the new enhanced LEACH as (upgraded LEACH, Enhanced LEACH)
b. Literature references, sufficient field background/context provided.
There is no citation to terminologies in their first appearance, as well there no citation the content prior the literature review section or after it even though it was needed. In addition, there is no consistency in the referencing format.
It seems that many of the references are quite dated, with most of them being from 2012. Additionally, it appears that the previous studies that have been reviewed only provided a surface-level analysis.
The literature review section needs to be rewritten.
c. Professional article structure, figures, tables. Raw data shared.
The structure of the article is in an acceptable format. Figures are relevant to the content of the article, Figures 6,7,8 resolution is not good some labels are not clear and the description in the captions is incomplete.
d. Self-contained with relevant results to hypotheses.
The new proposed LEACH should be compared to some of the studies reported in the literature review.
e. Formal results should include clear definitions of all terms and theorems, and detailed proofs.
The way results are plotted are not fully described and the results are not explained.

Experimental design

a. Original primary research within Aims and Scope of the journal.
YES.
b. Research question well defined, relevant & meaningful. It is stated how research fills an identified knowledge gap.
YES
c. Rigorous investigation performed to a high technical & ethical standard.
Yes.
d. Methods described with sufficient detail & information to replicate.
Not fully described.

Validity of the findings

a. Impact and novelty not assessed. Meaningful replication encouraged where rationale & benefit to literature is clearly stated.
The study is novel and a new algorithm is presented.
b. All underlying data have been provided; they are robust, statistically sound, & controlled.
Data is available.
c. Conclusions are well stated, linked to original research question & limited to supporting results.
YES.

---

## Round 0.2 · accepted · Accept

The paper has been improved. All reviewers' comments have been addressed. I recommend it for publication.

·

Basic reporting

Clear and unambiguous, professional English used throughout

Experimental design

Methods described with sufficient detail & information to replicate

Validity of the findings

Conclusions are well stated, linked to original research question & limited to supporting results